# Diversity of Herbicide-Resistance Mechanisms of *Avena fatua* L. to Acetyl-CoA Carboxylase-Inhibiting Herbicides in the Bajio, Mexico

**DOI:** 10.3390/plants11131644

**Published:** 2022-06-22

**Authors:** J Antonio Tafoya-Razo, Saul Alonso Mora-Munguía, Jesús R. Torres-García

**Affiliations:** 1Departamento de Parasitología Agrícola, Universidad Autónoma Chapingo, Texcoco 56230, Mexico; jtafoyar@chapingo.mx; 2I.T. Jiquilpan, Tecnológico Nacional de México, Jiquilpan de Juárez 59510, Mexico; l17420458@jiquilpan.tecnm.mx; 3Cátedras, Consejo Nacional de Ciencia y Tecnología, Mexico City 03940, Mexico; 4Laboratorio de Ecología y Evolución Molecular, Centro Interdisciplinario de Investigación para el Desarrollo Integral Regional (CIIDIR) del Instituto Politécnico Nacional, Unidad Michoacán, Jiquilpan de Juárez 59510, Mexico

**Keywords:** microevolution, adaptation, ecometabolomics, ACCase-inhibiting herbicides

## Abstract

Herbicide resistance is an evolutionary process that affects entire agricultural regions’ yield and productivity. The high number of farms and the diversity of weed management can generate hot selection spots throughout the regions. Resistant biotypes can present a diversity of mechanisms of resistance and resistance factors depending on selective conditions inside the farm; this situation is similar to predictions by the geographic mosaic theory of coevolution. In Mexico, the agricultural region of the Bajio has been affected by herbicide resistance for 25 years. To date, *Avena fatua* L. is one of the most abundant and problematic weed species. The objective of this study was to determine the mechanism of resistance of biotypes with failures in weed control in 70 wheat and barley crop fields in the Bajio, Mexico. The results showed that 70% of farms have biotypes with target site resistance (TSR). The most common mutations were Trp–1999–Cys, Asp–2078–Gly, Ile–2041–Asn, and some of such mutations confer cross-resistance to ACCase-inhibiting herbicides. Metabolomic fingerprinting showed four different metabolic expression patterns. The results confirmed that in the Bajio, there exist multiple selection sites for both resistance mechanisms, which proves that this area can be considered as a geographic mosaic of resistance.

## 1. Introduction

Herbicide resistance is an evolutionary process in response to the selection pressure that herbicides exert on weed populations that can negatively affect the yield and productivity of entire agricultural regions [1,2,3]. Since weed management can vary widely from farm to farm and each agricultural region has numerous farms (hundreds or even thousands), agricultural areas are essentially a selective mosaic with multiple possible hot spots of selection for resistance [2]. Weed management using herbicides generates a situation similar to that proposed in Thompson’s geographic mosaic theory of coevolution [4]. This selective factor (herbicides) can vary according to each farmer’s decisions (e.g., herbicide dose, the timing of application, and the number of applications, among others). Farmers can increase the dose or change the mode of action when the control is deemed deficient. Conversely, this intense selection pressure increases the presence of plant populations that have different and more efficient mechanisms of resistance, e.g., in the ACCase-inhibiting herbicides, the mutations in the site of action Ile–1781–Leu, Asp–2078–Gly, and Cys–2088–Ala confer cross-resistance for the three chemical families of such herbicides [3]. Additionally, this diversity in the management and high selection pressure can generate multiple hot spots of selection for resistant biotypes that evolved through different mechanisms (e.g., target-site resistance or nontarget-site resistance) in relatively small geographic scales [2,3,5,6].

In Mexico, the Bajio region (20°25′ N, 101°38′ W), one of the most important wheat and barley production zones, has reported problems with species-resistant ACCase-inhibiting herbicides since 1996 [7]. To date, resistant populations of *Avena fatua*, *Phalaris minor*, and *P. paradoxa* have been distributed in the Bajio. The first studies of the resistance mechanism in *A. fatua* and *P. minor* concluded that mutations in the site of action caused the resistance [8,9]. Among the main factors contributing to the high prevalence of herbicide resistance in the Bajio is farms’ large numbers and area. In this region, farm size ranges from 1 to 5 ha, and there are about 24,000 farms, where the application of management practices depends upon each farmer’s decisions. Failures in weed management, such as selecting the wrong type of herbicide to apply, applying at too-low doses, applying too late, and using generic products, increase the risk of selection for resistant biotypes.

Attempts to eradicate these resistant populations include the introduction of pinoxaden. This new herbicide showed excellent weed control during the following two years after implementation. However, a high number of resistant populations have surged in the Bajio. Some samplings show that in most of them, the cause of resistance is a mutation in the site of action. However, in around 20% of farms sampled, genetic analyses found no mutations at the site of action. Therefore, it is possible that populations present nontarget-site resistance (NTSR) based on the overexpression of cytochrome P450. For this reason, it is possible that in the Bajio, there coexist different mechanisms of resistance (genetic and metabolic) in response to selective pressure imposed by herbicides.

In the case of herbicide resistance caused by mutations in the site of action, the resistant biotypes can be identified and studied in great detail using molecular markers [2,10,11,12]. However, in the case of metabolic resistance based on overexpression of genes from the cytochrome P450 family of genes (hereafter, P450), it is not possible to differentiate among biotypes using molecular markers because the difference between R and S populations is the overexpression of specific genes [13,14]. In this sense, the use of metabolomics techniques can be helpful to obtain a global pattern of expression (metabolic fingerprinting) in resistant populations and to establish if two or more biotypes share the same detoxification pathways or belong to different populations [15].

It is important to know the abundance of each mechanism of action mutation that confers herbicide resistance, which thus can give information about the selection pressure that herbicides exert in the region. For example, the increase in mutations that confer the same cross-resistance as Ile–1781–Leu, Asp–2078–Gly, and Cys–2088–Ala can indicate the high selection pressure imposed by herbicides. In this study, we hypothesized that due to the complexity of weed management in the Bajio, it is possible to find many resistant biotypes of *A. fatua* based on TSR and NTSR that have evolved independently to resistance. This could confirm that this agricultural region showed the predictions of the geographic mosaic theory. Therefore, the objectives of this study were: (1) to sequence the site of action of *A. fatua* populations with failures in weed control; (2) to apply a P450 gene inhibitor to determine if, in populations without mutations in the site of action, the cause of the resistance is the overexpression of P450 genes; (3) to use metabolomic fingerprinting to differentiate among biotypes with metabolic resistance based on their possible detoxification pathway.

## 2. Results

### 2.1. Sequence Analyses

The sequencing analyses of the two domains of the *ACCase* gene that include all known mutations conferring resistance to ACCase-inhibiting herbicides were made using 802 nucleotides, which were translated to 267 amino acids. A total of 49 different biotypes were detected with mutations in the site of action, representing 70% of the total accessions sampled.

Figure 1 shows the distribution of the different biotypes along the Bajio. All biotypes were distributed in a random pattern. No evidence of structure in the populations was observed. Besides the high number of biotypes, all mutants reported that confer resistance to ACCase-inhibiting herbicides were found in this study. The most abundant biotype was the mutant Trp–1999–Cys with 16 records; conversely, the less abundant biotype was the mutant Gly–2096–Ala with one record. Another important issue to consider is the presence of more than one biotype inside the crop fields. The results showed that four field crops had the presence of two different mutants (Ile–1781–Leu/Ile–2041–Asn, Ile–1781–Leu/Gly–2096–Ala, Trp–2027–Cys/Asp–2078–Gly, Asp–2078–Gly/Cys–2088–Arg).

According to the level of resistance conferred by the type of mutation, the mutants Ile–1781–Leu, Asp–2078–Gly, and Cys–2088–Arg were resistant to the three chemical families ACCase-inhibiting herbicides (cross-resistance) and represent 43% of the biotypes with TSR. The mutants Trp–2027–Cys and Ile–2041–Asn are considered medium resistant and present in 22% of the accessions. Finally, the last 36% are confirmed by the mutants Trp–1999–Cys, Ile–2041–Val, and Gly–2096–Ala, which show reduced resistance to ACCase-inhibiting herbicides (Figure 2).

A total of 21 accessions did not show mutations in the site of action that explain their lack of control with herbicides. Of these accessions, six were selected for corroboration of resistance by a dose–response test and with the P450 inhibitor, followed by metabolic fingerprinting.

### 2.2. Evaluation of Resistance of Biotypes without Mutations in the Site of Action

The application of clodinafop–propargyl (an ACCase-inhibiting herbicide) to individuals from populations collected in the Bajio showed that all populations had some degree of resistance (Figure 3A), although the populations varied in their RI (resistance index) from 1.43 for Manuel Doblado to 7.07 for Abasolo (Table 1). The populations from Cueramaro, Presidio, Providencia, and Yuriria had similar RIs, around 3.5. When malathion was applied prior to the herbicide application, RI decreased in all populations to a level of sensitivity similar to susceptible biotypes (Figure 3B).

### 2.3. Metabolic Fingerprinting of Populations Using DIESI-MS

A total of 1460 metabolites were detected using DIESI-MS for metabolomic fingerprinting, of which 399 were negatively charged, and 1061 were positively charged. The resulting heat map was able to separate treatments (control and herbicide application) and populations according to responses to herbicide application.

The superior dendrogram shows the formation of two main branches. The first branch (right side of the heat map) is composed of all treatments under the control condition. In this branch, four subgroups were formed: Abasolo and Yuriria, Cueramaro, Manuel Doblado, and Presidio and Providencia (Figure 4). However, the metabolic fingerprinting changed significantly when herbicide was applied, causing the grouping of these treatments in the other branch (left side of the heat map). In addition to conforming to a separate branch from the control treatments, the populations were grouped differently within this branch, forming four different subgroups: Presidio, Manuel Doblado and Providencia, Abasolo and Cueramaro, and Yuriria.

According to their metabolomic fingerprinting and their respective branch separation within the herbicide-treated plants, we established that the populations in the same subgroups belonged to the same biotype since we assumed that the pattern of metabolite expression is related to the detoxification pathways that each biotype has for the degradation of herbicide. In this biotype assignation, the biotype Presidio was the most distant among the other biotypes, while Manuel Doblado and Providencia constituted the biotype with the highest similarity in metabolism.

The lateral branch (left side of the heat map) shows the grouping of the expression of metabolites (Figure 4). Within the heat map, there are zones where the ionization of metabolites changed with herbicide application, and the intensity of each metabolite was useful to determine the relationships among populations (quantitative and qualitative changes).

However, the grouping of biotypes did not correspond to the geographic distance among populations. For example, Cueramaro and Abasolo belonged to the same biotype and were separated by 25 km, and the Manuel Doblado and Providencia populations were highly correlated but were separated by 82 km.

## 3. Discussion

The agricultural area of the Bajio showed a higher number of *A. fatua* biotypes resistant to ACCase-inhibiting herbicides. Both mechanisms for resistance (TSR and NTSR) were detected, and all mutations reported to confer herbicide resistance to ACCase-inhibiting herbicides are present in the Bajio. Furthermore, metabolomic fingerprinting can detect at least four distinct metabolite expression patterns in biotypes with resistance by overexpression of P450. This evidence confirms that in the Bajio, there exists a high number of hot spots caused by the agricultural practices that increase the selection of resistant biotypes. We can assume that agricultural areas can be studied as geographic mosaics of coevolution. In this zone, it is possible to study events of evolutionary interest such as microevolution, local adaptation, convergent evolution in genetics and metabolic pathways, and evolutionary physiology, among others [16,17].

In the Bajio, there has been a history of herbicide resistance since 1996 [7]. The conditions shown in this research confirm that the high number of farms and the heterogeneity of weed management is the cause of the selection of resistant biotypes in almost three species. Each one of the species with resistant biotypes had a diversity of mutants. In *P. minor*, the biology and evolutionary origin of resistance in at least four mutants (Ile–1781–Leu, Trp–2027–Cys, Ile–2041–Asn, and Asp–2078–Gly) have been described. In addition, recent studies have also described the presence of biotypes of *Echinochloa crus-galli* [18] resistant to ALS-inhibiting herbicides and a biotype of *A. fatua* with multiple resistance to ALS and ACCase-inhibiting herbicides [19]. However, this is the first report describing TSR and NTSR mechanisms for this zone.

All populations of A. fatua sampled in this study from the Bajio, Mexico, were resistant to ACCase-inhibiting herbicides. However, the sampling collection was focused on farms with evident fails in weed control. For this reason, the high amount of resistant biotypes does not indicate that all farms in the Bajio are affected by the presence of resistant biotypes. Although, in the Bajio, the principal mechanism of herbicide resistance is the TSR; around 70% of the total biotypes collected had mutations in the site of action. Furthermore, all mutations reported conferring resistance to ACCase-inhibiting herbicides are present in the Bajio.

A similar situation occurs in the European populations of *Alopecurus myosuroides* (black-grass) in France, the United Kingdom, Belgium, the Netherlands, Germany, and Turkey, where TSR represents two-thirds of resistant biotypes to ACCase inhibiting-herbicides. [10] However, in the case of the Bajio, the geographical extension is many folds less than such European regions.

The high diversity of mutations is caused principally by the independent origins of evolution. Studies made in different agricultural regions with herbicide resistance in other parts of the world confirm that resistance can arise from multiple independent origins of evolution and present the two resistance mechanisms (TSR and NTSR). For example, in France, *Ambrosia artemisiifolia* L. is resistant to ALS-inhibiting herbicides and is distributed throughout the country. Population genetics analyses showed that many biotypes evolved independently to resistance [20]. The same situation of multiple independent origins of evolution is present in *Amaranthus tuberculatus* populations resistant to ALS-inhibiting herbicides in the United States of America. In addition, widespread gene flow also increases the complexity of the management [21].

According to the diversity of mutations in the site of action, it is possible to establish that resistance has evolved at least eight times in this region (one for each mutation). It is possible that many biotypes with the same mutation had independent evolutionary origins and, for evolutionary convergence, share the same mutation. However, in practice, it is difficult to determine if biotypes with the same mutation result from independent selective events or are derived from the same population and are now dispersed in the Bajio. The evidence that dispersion is also a factor in the Bajio is the presence of two distinct mutants on the same farm. The most feasible explanation is the dispersion of seeds across the Bajio. The probability that two different mutations evolved to resistance in the same farm is low.

This random distribution of biotypes and the lack of genetic structure in the populations are frequent in other agricultural regions [22]; the dispersion increases the complexity of the problem. For this reason, the distribution of herbicide-resistant populations is considered similar to epidemiological cases [23]. However, the most common situation in epidemiology is a unique origin and posterior dispersion. Therefore, we propose conceptualizing the agricultural areas as selective mosaics caused by the differences in weed management.

The number of biotypes that present cross-resistance (Ile–1781–Leu, Asp–2078–Gly, Cys–2088–Arg) represents 43% of the total biotypes with a mutation in the site of action. It is possible that the introduction of pinoxaden increases the abundance of such biotypes and decreases mutations that confer lower levels of resistance, such as Trp–1999–Cys, Ile–2041–Val, Gly–2096–Ala. The introduction or discovery of new molecules is not visualized in the next five years [24]. For this reason, if this tendency continues, the possibility of weed control in the Bajio will be limited.

Conversely, the NTSR was confirmed in biotypes that did not show mutations in the site of action. Doses–response tests confirmed the resistance, and the application of malathion showed that P450 genes are involved in herbicide detoxification, eliminating the possibility of another mechanism being the cause of reduced translocation of the herbicide, multiple copies of the gene in the genome, or overexpression of the site of action, among others [25,26,27]. Metabolomic fingerprinting using DIESI-MS was able to identify 1460 metabolites, allowing for the identification of similarities and differences in metabolic expression that allowed the populations to be grouped by biotype according to different possible pathways of herbicide detoxification. Because resistance in the studied biotypes was caused by enhanced metabolism to degrade herbicides, classification according to metabolomic fingerprinting could be an approach for detecting differences among populations and for assigning biotypes. To date, the use of mass spectrometry methods has not been employed in the study of herbicide-resistant populations in the field. In the case of herbicide-resistant weeds, Torres-García et al. [19] used this method to determine the differences in metabolic fingerprinting before and after the herbicide applications in a multiple-resistant A. fatua biotype.

In practice, the causes of the selection of TSR and NTSR are different. While the intense selection pressure that herbicides exert on populations (mortality > 99.9%) causes TSR, NTSR is caused by failures in the application of the herbicides, specifically by low doses [28,29,30]. This can occur because the herbicide is applied too late (weeds >10 cm tall), by the use of generic products (with a lower concentration of active ingredients), or by the intentional reduction of doses by farmers in an attempt to save money by buying less herbicide. The main problem with metabolic resistance is that the resistant individuals can develop resistance to other herbicides to which they have never been exposed [31].

The presence of many biotypes in the Bajio based on TSR and NTSR is evidenced of the high selection pressure in the zone for more than 25 years. The high number of farms, weed management diversity, and the changes in chemical families of ACCase-inhibiting herbicides have contributed to the selection of resistant phenotypes better adapted to crop conditions. In this sense, the comparison of agricultural regions with herbicide resistance with respect to geographic mosaics of selection is congruent. Weed populations respond to selection pressures imposed by farming practices and develop adaptive mechanisms to survive the herbicide applications. Therefore, the management practices to reduce the impact of herbicide resistance in the zone should be based on the individual management of each farm. Generalized weed management is not recommended because each farm is a hot spot where the resistant biotypes were selected. Each biotype is locally adapted to its field, and biological traits are synchronized to management practices such as the dates of sowing, physiological maturity, and tillage, among others [9].

## 4. Methods

### 4.1. Collecting of Biotypes

The leaf plant tissue and seeds collection was carried out in the southwestern portion of Mexico’s Bajio region. In recent years, this region has increased the number of reports of failures in weed control. We collected leaves and seeds from 70 farms where chemical control is the only mode of weed control and where the presence of resistant individuals is evident (null or very deficient control of weeds). The name of each accession was established according to the name of the municipality where it was collected (Figure 1).

Samples were collected inside wheat fields during the 2021 growing season. In addition, plant tissue samples were collected throughout the farm, avoiding edges where weeds may be missed in herbicide applications. The samples were introduced in plastic bags and stored in liquid nitrogen until their use. Only physiologically mature plants were chosen for seed collection to ensure that seeds would be completely developed. The collected seeds were placed in paper bags and maintained in refrigeration (4 °C) until their use.

### 4.2. Sequencing the Site of Action

DNA extraction was performed by the CTAB method using approximately 100 mg of plant tissues [32]. For each population, a total of five individuals were used. After the extraction, the quality of DNA samples was evaluated by a NanoDropTM instrument (Thermo Scientific, Waltham, MA, USA); only samples with a quality between 1.8 and 2 were used (the result of the reading 260/288 nm). Samples were diluted to a concentration of 50 ng μL^−1^.

To determine whether mutations at the site of action conferred herbicide resistance in the populations studied, PCR amplification of two regions of the *ACCase* gene was carried out (from position 5109 to 5632 and from 5950 to 6355) using two sets of universal primers developed by Délye and Michel [33] (Table 2). These primers amplify two fragments of the *ACCase* gene where all known resistance mutations occur: in the codons 1781, 1999, 2027, 2041, 2078, 2088, and 2096. The PCR mix contained 1X of the buffer, 10 mM of dNTPs, 1 µL of DNA, 1.5 mM of MgCl2, 1 U of Taq (Flexitaq; Promega), and 10 mM of each primer in a final volume of 25 µL. The amplification conditions were as follows: denaturation at 95 °C for 30 s followed by 37 cycles of 95 °C for 10 s, 60 or 61 °C (depending on the fragment) for 15 s, and 72 °C for 45 s, followed by a final extension at 72 °C for 10 min. The amplicons were sent to Macrogen, Korea, for sequencing.

### 4.3. Sequence Analyses

Sequence analysis was performed using the software MEGA (Molecular Evolutionary Genetics Analysis) version 7 [34]. Before analysis, we confirmed the quality and effective size of the sequences included in the alignment. Then, the sequences were aligned, and the translation to amino acids was accomplished using the chloroplast genetic code. For the *ACCase* gene, the two fragments were linked, indicating each fragment’s start and end site. The alignment of each domain included an *A. fatua* sequence reported in GenBank as susceptible to ACCase-inhibiting herbicides (KJ606970.1). Special attention was paid to the non-synonymous mutations reported as responsible for resistance to ACCase-inhibiting herbicides (Ile–1781–Leu, Trp–1999–Cys, Trp–2027–Cys, Ile–2041–Asn, Ile–2041–Val, Asp–2078–Gly, Cys–2088–Arg, Gly–2096–Ala) [3].

### 4.4. Evaluation of Resistance

When the populations did not show mutations in the site of action that confer herbicide resistance, dose–response tests were performed to corroborate the resistance. This test was carried out in a greenhouse in the department of agricultural parasitology at the Universidad Autónoma Chapingo. The temperature was maintained between 16 and 28 °C, and humidity was regulated. Four-liter pots were filled with *Sphagnum* peat moss (Miracle-gro), and about 30 *A. fatua* seeds were sown. After the emergence of seedlings, the seedlings were culled, leaving ten plants in each pot. Herbicide was applied when the seedlings were 10 cm tall. Seven different doses of clodinafop–propargyl (Topik Gold^®^, Syngenta) were tested: 0, 1.87, 3.75, 7.5, 15, 30, 60, 120, and 240 g a. i. ha^−1^, corresponding to 0, 1/32, 1/16, 1/8, 1/4, 1/2, 1, 2 and 4 times the recommended field doses, respectively. Two mL of mineral oil per L of the solution was added to the herbicide solution as an adjuvant. The herbicide was applied using a pressurized CO_2_ plot sprayer calibrated to a spray volume of 200 L per hectare. Five replicates were carried out for each dose. Twenty days after herbicide application, the percentage of biomass reduction (dry matter) at each dose was compared to that of the untreated plants. Plant shoots were harvested and placed in paper bags, and then the plants were dried for 72 h at 80 °C until reaching constant weight. The dose required to reduce shoot weight by 50% relative to untreated plants (GR_50_) for each population was calculated using the logistic model using the statistical software SigmaPlot 11 (Equation (1), where C = lower limit, D = upper limit, b = slope, and GR_50_ = dose yielding 50% biomass reduction) [35]. The resistance index (RI) was calculated by dividing the GR_50_ of the resistant biotype by the GR_50_ of the susceptible (control) biotype.
y = C + {D − C/1 + exp[b(logx − logGR_50_)]}(1)

A second experiment was carried out to determine whether the resistance mechanism was the overexpression of P450 genes. In this experiment, malathion (an inhibitor of the synthesis of P450 genes) was applied at a dose of 1× one hour before herbicide application [25]. The protocol for application, evaluation of the dry matter, and calculation of the resistance index were the same as described above.

### 4.5. Metabolic Fingerprinting of Populations Using DIESI-MS

To determine whether the populations with metabolic resistance shared the same metabolite expression, metabolic fingerprinting of each population was obtained under control conditions (without the herbicide application, only adjuvant) and with the herbicide application. The method used for metabolic fingerprinting was DIESI-MS. In previous studies, we have implemented such a method to determine the global fingerprinting of a multiple-resistant biotype when sprayed with herbicides (ACCase- and ALS-inhibiting herbicides) [19]. Nevertheless, this method has not been standardized and calibrated to determine metabolic pathways in herbicide detoxification. However, this method is useful in classifying and grouping samples according to metabolite expression [36,37]. For this reason, this method only was used to establish if populations with resistance based on the overexpression of P450 shared the same metabolite expression. If the biotypes show a similar pattern in the expression of metabolites, it would suggest that both populations share the same metabolic pathways of detoxification and that both populations belong to the same biotype and came from the same selective event for resistance.

The germination and seedling growth protocols were the same as the dose–response test. Herbicides were applied when the seedlings were 10 cm tall. The treatments consisted of a control (applying only the adjuvant), and the application of 1x the recommended dose of clodinafop–propargyl. Plant shoots were collected 24 h after herbicide application (eight plants per treatment). Samples were washed with distilled water for 5 min to eliminate herbicide residue from the leaves. They were placed in paper bags and flash-frozen in liquid nitrogen to stop metabolism. The tissue was lyophilized in a vacuum at −50 °C for 96 h. The dry samples were ground in a metal mill (Retsch MM 400) to a fine powder (~5 μm). Then, 100 mg of powder was extracted with 1 mL 80% methanol at room temperature (Mass grade, Fisher Scientific). Samples were homogenized for 20 min in a sonifier (Branson 1800) to improve metabolite extraction at 4 °C. Extracts were centrifuged at 13,000 rpm for 10 min at 4 °C. This suspension was passed through a 0.2 µm nylon filter (Millipore).

The injection conditions and data analysis have been described previously in Torres-García [19]. The main features of the script used are spectral merger to improve accuracy, curve smoothing to increase signal-to-noise ratio, ion binning by rounding of the masses of the peaks into ~1 Da intervals, Tukey test among treatments for each metabolite, boxplot analysis for matrix construction (using median values) and representation by heatmap-bicluster that builds dendrograms using Pearson correlation as a distance function [38]. For constructing the heat map, an ion matrix was constructed using the 100 metabolites with the highest ionization and using log-transformed data and R correlation as a distance for clustering [39].

## 5. Conclusions

The evidence indicates that the Bajio is an area with a great diversity of weed management and is correlated with the selection of different resistance mechanisms. The principal resistant mechanism is TSR, where there exist all the mutations reported for ACCase-inhibiting herbicides. Moreover, the selection pressure imposed by pinoxaden increases the abundance of mutations that confers cross-resistance and the selection of biotypes with NTSR. Finally, even in biotypes with NTSR, differences exist in the metabolomic fingerprinting of some accessions; this suggests that NTSR has multiple independent origins of evolution. For these reasons, we conclude that the Bajio is a selective mosaic for herbicide resistance selection.

## Figures and Tables

**Figure 1 plants-11-01644-f001:**
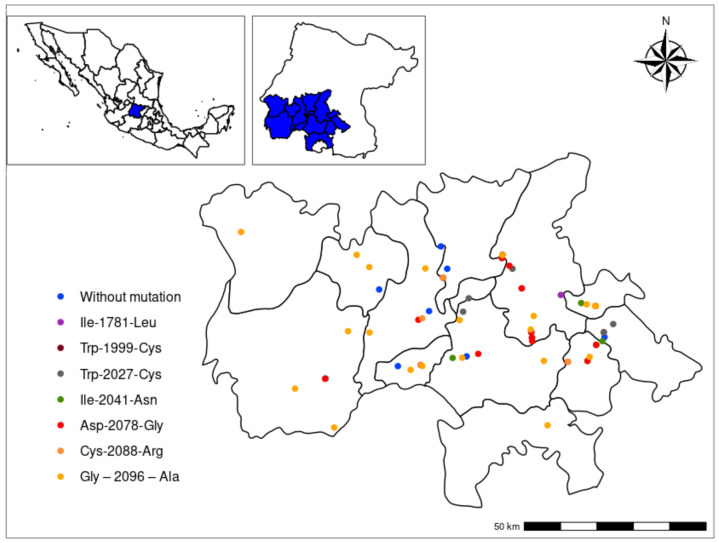
Map of the distribution of different *Avena fatua* L. biotypes herbicide resistant to ACCase-inhibiting herbicides in the Bajio, Mexico.

**Figure 2 plants-11-01644-f002:**
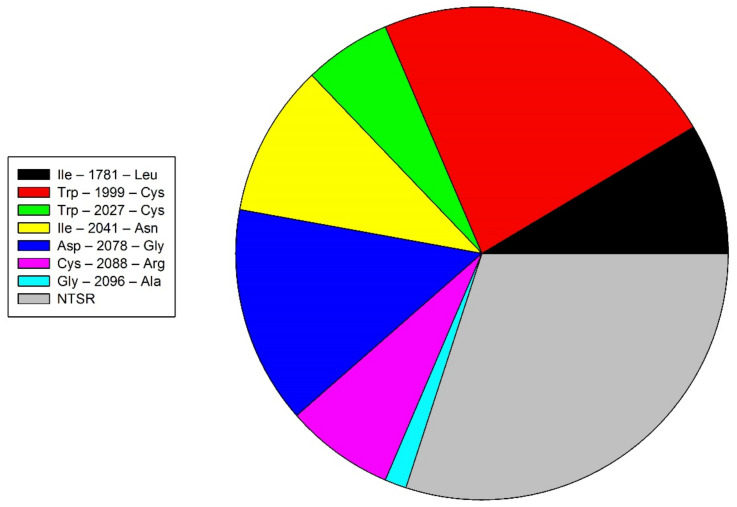
Proportion of mechanisms of action of *Avena fatua* L. populations resistant to ACCase-inhibiting herbicides in the Bajio, Mexico.

**Figure 3 plants-11-01644-f003:**
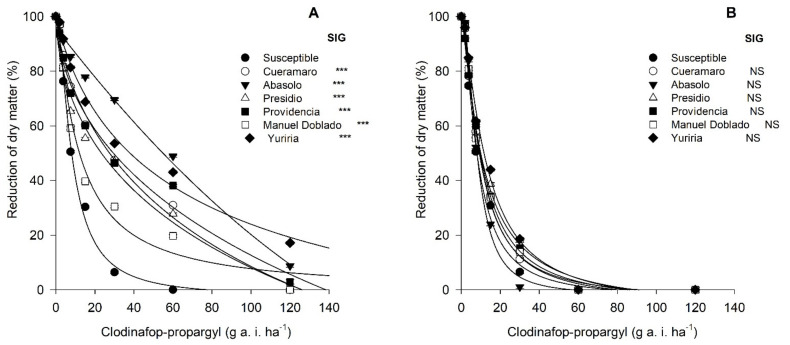
Dose–response tests of biotypes of *Avena fatua* L. resistant to ACCase-inhibiting from the Bajio, Mexico. In (**A**), the test was made only with clodinafop–propargyl (0, 1.87, 3.75, 7.5, 15, 30, 60, 120, and 240 g a. i. ha^−1^). In (**B**), malathion was applied one hour before the herbicide application. Asterisk indicates significant statistical (*p* ≤ 0.001) with respect to susceptible biotype, NS indicates non-significant differences.

**Figure 4 plants-11-01644-f004:**
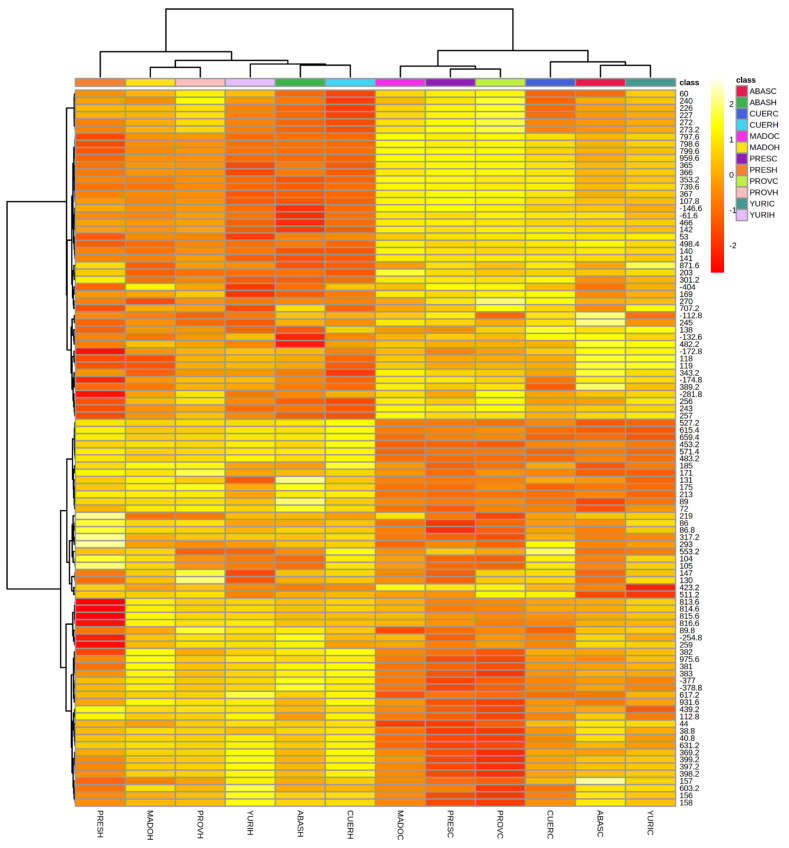
Metabolic fingerprint of populations of *Avena fatua* L. resistant to ACCase-inhibiting herbicides from the Bajio, Mexico. The heatmap is constructed with 100 metabolites with the highest ionization. The colors represent the abundance of the metabolites. The ions are clustered according to their R correlation (left dendrogram), while the treatments are also clustered according to their R correlation (top dendrogram). The first four letters indicate the name of the populations: Abasolo (ABAS), Yuriria (YURI), Cueramaro (CUER), Manuel Doblado (MADO), Presidio (PRES) and Providencia (PROV). Treatments with herbicides are indicated with the letter C (control conditions) and H (herbicide application).

**Table 1 plants-11-01644-t001:** Growth reduction at 50% (GR_50_) and resistance index (RI) of populations of *Avena fatua* L. with the application of ACCase-inhibiting herbicide with and without prior malathion application.

Population	GR_50_	RI	GR_50_	RI
	without the Application of Malathion		with the Application of Malathion	
Susceptible	7.33		7.295	
Cueramaro	26.2	3.57	9.67	1.32
Abasolo	51.83	7.07	7.93	1.08
Manuel Doblado	10.55	1.43	9.05	1.24
Presidio	23.7	3.23	10.07	1.38
Providencia	23.99	3.27	10.29	1.41
Yuriria	24.6	3.35	12.25	1.68

**Table 2 plants-11-01644-t002:** Name of primer, sequence, fragment size (bp), and annealing temperature (°C) of the primers that were used to amplify the ACCase regions in *Avena fatua* [33].

Primer	Sequence (5′–3′)	Fragment Size	Annealing Temperature (°C)
ACcp1	CAACTCTGGTGCTIGGATIGGCA	523	60
ACcp1R	GAACATAICTGAGCCACCTIAATATATT		
ACcp4	CAGCITGATTCCCAIGAGCGITC	405	61
ACcp2R	CCATGCAITCTTIGAGITCCTCTGA		

## Data Availability

All data generated and analyzed in this study are included in this paper.

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
