# Peer review of "Diversity of Herbicide-Resistance Mechanisms of Avena fatua L. to Acetyl-CoA Carboxylase-Inhibiting Herbicides in the Bajio, Mexico"

_plants, 2022, doi:10.3390/plants11131644_

Round 1

Reviewer 1 Report

The current work, Diversity of herbicide-resistance mechanisms of Avena fatua L. to ACCase-inhibiting herbicides in the Bajio, Mexico fits within the scope of the journal Plants. The results of presented study can be considered of interest in order to achieve more effectively farming due to the individual management of herbicide applications in agricultural regions with different herbicide resistance.

The paper needs however some revisions. Please reexamine the paper taking into account the following:

1. Title: I suggest in Title to write the full name of AACase – Acetyl-CoA carboxylase.

2. Abstract: Please check and avoid using abbreviation in Abstract without their definition (for example, TSR).

3. Introduction: L30: “The Introduction” in the beginning of sentence should be deleted

4. Results:

4.1. L133: Please add the full name of abbreviation RI by the first mention.

4.2. Figure 4: the numbers on the right are not readable. If possible, please revised the figure.

5. Discussion: L216-224: The information in this paragraph is mostly duplicate the information from Introduction. Please revise this paragraph to avoid the overlapping.

6. Please check the Instruction for authors: The numbers of references should be placed before point at the end of sentences.

Author Response

1. Title: I suggest in Title to write the full name of AACase – Acetyl-CoA carboxylase.

R= Done

2. Abstract: Please check and avoid using abbreviation in Abstract without their definition (for example, TSR).

R= Done

3. Introduction: L30: “The Introduction” in the beginning of sentence should be deleted

R= Done

4. Results:

4.1. L133: Please add the full name of abbreviation RI by the first mention.

R= Done

4.2. Figure 4: the numbers on the right are not readable. If possible, please revised the figure.

R= Done. The figure was improved. A new figure was constructed using the same statistical treatment. The pallet color was changed, and the numbers are better defined. The only difference between figures is the position of the treatments. In the new heat map treatments with herbicide application are in the left side.

5. Discussion: L216-224: The information in this paragraph is mostly duplicate the information from Introduction. Please revise this paragraph to avoid the overlapping.

R= the paragraph in the discussion was changed

6. Please check the Instruction for authors: The numbers of references should be placed before point at the end of sentences.

R= Done

Reviewer 2 Report

The discussed manuscript discusses the topic of weed resistance to herbicides, which is important not only in Mexico, but also around the world. Before publishing it, it would be worth paying attention to a few elements and supplementing them.

  1. There are no conclusions or any summary of the research carried out in the study.
  2. The authors cite most of the works from earlier years, i.e. 2018 and earlier. Only one of the cited works was published in 2020. In this most recent period, has there been no published paper on herbicide resistance that is noteworthy and provides new insight?
  3. The authors focused closely on discussing weed resistance in Mexico's Bajo region. Wouldn't it be worth extending some discussion of the obtained results by comparing the obtained results to the results presented by other scientists in the world? Avena fatua L. is not the only weed to show resistance to herbicides and Mexico is not the only place where it occurs.
  4. I suggest you read the work carefully in order to eliminate minor errors, e.g. line 30, page 1.

Author Response

  1. There are no conclusions or any summary of the research carried out in the study.

    R= Conclusions were added.

  2. The authors cite most of the works from earlier years, i.e. 2018 and earlier. Only one of the cited works was published in 2020. In this most recent period, has there been no published paper on herbicide resistance that is noteworthy and provides new insight?

  3. R=Four new cites were added.

    Dixon, A., Comont, D., Gancho, T. S., Neve, P. Population genomics of selectively neutral genetic structure and herbicide resistance in UK populations of Alopecurus myosuroides. Pest Manag. Sci. 2021 77, 1520–1529.

    Comont, D., Neve, P. Adopting epidemiological approaches for herbicide resistance monitoring and management Weed Res. 2021 61, 81–87.

    Julia M Kreiner, J. M., Sandler, G., J Stern, A. J., Tranel, P. J., Weigel, D., Stinchcombe, J. R., Wright, S. I. Repeated origins, widespread gene flow, and allelic interactions of target-site herbicide resistance mutations. ELife 2022 11, e70242. DOI: https://doi.org/10.7554/eLife.70242

    Loubet, I., Caddoux, L., Fontaine, S., Michel, S., Pernin, F., Barrès, B., Le Corre, V., Délye, C. A high diversity of mechanisms endows ALS-inhibiting herbicide resistance in the invasive common ragweed (Ambrosia artemisiifolia L.). Sci. Rep 2021 11, 19904 doi.org/10.1038/s41598-021-99306-9

  4. The authors focused closely on discussing weed resistance in Mexico's Bajo region. Wouldn't it be worth extending some discussion of the obtained results by comparing the obtained results to the results presented by other scientists in the world? Avena fatua L. is not the only weed to show resistance to herbicides and Mexico is not the only place where it occurs.

    R= in the discussion, a perspective of the geographic mosaic of selection was emphasized to obtain a global perspective the discussion and avoid a closely discussing

Round 2

Reviewer 2 Report

The article can be accepted in its present form